# Prolonged Stress Causes Depression in Frontline Workers Facing the COVID-19 Pandemic—A Repeated Cross-Sectional Study in a COVID-19 Hub-Hospital in Central Italy

**DOI:** 10.3390/ijerph18147316

**Published:** 2021-07-08

**Authors:** Nicola Magnavita, Paolo Maurizio Soave, Massimo Antonelli

**Affiliations:** 1Postgraduate School of Occupational Medicine, Università Cattolica del Sacro Cuore, 00168 Rome, Italy; paolomaurizio.soave@policlinicogemelli.it; 2Department of Woman/Child & Public Health, Fondazione Policlinico Universitario Agostino Gemelli IRCCS, 00168 Rome, Italy; 3Department of Emergency, Anesthesiology and Resuscitation Sciences, Fondazione Policlinico Universitario Agostino Gemelli IRCCS, 00168 Rome, Italy; massimo.antonelli@unicatt.it

**Keywords:** emergency, infectious disease, organizational justice, stress, loneliness, compassion fatigue, meditation, prayer, insomnia, mental health, perspective study

## Abstract

The COVID-19 pandemic has severely tested the mental health of frontline health care workers. A repeated cross-sectional study can provide information on how their mental health evolved during the various phases of the pandemic. The intensivists of a COVID-19 hub hospital in Rome were investigated with a baseline survey during the first wave of the pandemic in April 2020, and they were contacted again in December 2020, during the second wave. Of the 205 eligible workers, 152 responded to an online questionnaire designed to measure procedural justice, occupational stress (effort/reward imbalance), sleep quality, anxiety, depression, burnout, job satisfaction, happiness, and turnover intention. Workers reported a further increase in workload and compassion fatigue, which had already risen during the first wave, and a marked reduction in the time devoted to meditation and mental activities. A low level of confidence in the adequacy of safety procedures and the need to work in isolation, together with an increased workload and lack of time for meditation, were the most significant predictors of occupational stress in a stepwise linear regression model. Occupational stress was, in turn, a significant predictor of insomnia, anxiety, low job satisfaction, burnout, and intention to leave the hospital. The number of workers manifesting symptoms of depression increased significantly to exceed 60%. Action to prevent occupational risks and enhance individual resilience cannot be postponed.

## 1. Introduction

The pandemic of coronavirus disease 2019 (COVID-19), caused by severe acute respiratory syndrome coronavirus 2 (SARS-CoV-2), has dramatically tested health services all over the world. Since being hit by the first wave of the epidemic in the spring of 2020 [1] and the second wave in the autumn of the same year [2], Italy has been one of the countries most affected. For health care workers (HCWs) the two waves posed different problems. In the first phase of the outbreak, the sudden overload of work, the lack of protective equipment, fear of infection, insufficient knowledge of safety procedures, and uncertainty about treatment criteria were among the major problems [3,4]. In the second phase, once the shortage of devices had been resolved, the new safety procedures had been assimilated, and the therapeutic protocols had been consolidated, the psychosocial problems related to the on-going epidemic became evident. At the same time, public opinion towards HCWs was beginning to rapidly move in a more negative direction [5].

Since the early months of the COVID-19 epidemic, numerous scientific papers have considered the possibility that frontline HCWs are being affected by post-traumatic stress, anxiety, depression, and burnout. During epidemics, intensivists are among the most vulnerable HCWs on account of infections and mental health problems [6]. Residents have the greatest difficulties in adapting to the new security measures [7]. After Lai et al. [8] estimated the prevalence of distress in Chinese HCWs to be 71.5%, extensive research has also led to a large number of systematic reviews and meta-analyses [9,10,11,12,13] that have confirmed a high prevalence of occupational distress and mental health problems in HCWs treating COVID-19 patients. In frontline HCWs, the pooled prevalence of anxiety has been estimated to range from 23.2% [11] to 32.0% [10] and that of depression from 22.8% [11] to 28% [10]. However, all the studies included in these reviews were cross-sectional or, at best, retrospective. Psychic symptoms in HCWs were generally compared to “normal values”, administrative staff, or external samples; moreover, many studies lacked a control group [14]. Overall, a lack of pre-/post-longitudinal studies made it impossible to distinguish the effect of the pandemic from that of all other pre-existing stressors in hospital work. Only exceptionally, in fact, did researchers have mental health data collected before the pandemic to compare with those recorded during the pandemic. For example, Kok et al. compared data on the prevalence of burnout before of the pandemic with those recorded during the pandemic peak in ICU personnel from a university medical center, having a 50% response rate at follow-up. The study indicated that overburdening of HCWs during an extended period of time had increased the symptoms of burnout [15]. Burnout, however, is a morbid condition that occurs after exposure to chronic stress and is therefore not the first indicator of mental health that can be altered by an epidemic [16]. There are conflicting findings on the epidemiology of burnout among HCWs working in COVID-19 wards: some studies have found a reduction in burnout rates, for example in US neurosurgeons [17], in French geriatric facilities [18], and in Chinese frontline nurses, compared with ordinary ward workers [19]. The mental health of HCWs can vary according to many factors and is hardly homogeneous for all groups of workers, even in the same company or department. A survey conducted in March–April 2020, on the staff of an Italian local health unit, showed that the levels of occupational stress, anxiety, and depression during the first phase of the epidemic were on average no higher than those recorded over the years in the cohort. However, HCWs who had unprotected contact with COVID-19 patients and even more those who had SARS-CoV-2 positive nasopharyngeal swab had an increased risk of insomnia, anxiety, and depression compared to their colleagues [4].

As noted, in the absence of pre-/post-studies, further longitudinal study is needed to distinguish psychological symptoms during and after the infectious disease outbreaks [20]. With the onset of the pandemic, many researchers have carried out short-term longitudinal studies (one to three months) on specific topics relating to the mental health of HCWs. The studies published so far have been conducted online and anonymously, without the possibility of tracking respondents; consequently, the surveys carried out at different moments of the pandemic do not represent longitudinal studies, in which the incidence of mental disorders can be evaluated, but repeated cross-sectional studies, with comparison of prevalence and mean values. All these studies have a very short duration, from one to three months, and have had a very low response rate: the number of participants was always much smaller than the total achievable population, and participation decreased very rapidly over time. For example, Sasaki et al. performed a prospective online cohort study of a population of 4120 full-time employees, measuring psychological distress in the last 30 days by the Brief Job Stress Questionnaire. The two-months longitudinal study recruited a mixed population of 996 people, 111 of whom were HCWs [21]; the participation dropped to 83 HCWs vs 863 controls in the eighth month [22]; on both occasions the HCWs had a higher mean stress score than the controls. Several research groups have observed that stress and anxiety levels tend to decrease in the transition between the attack and the stationary phase of the pandemic. Chew et al. performed a three-months repeated survey of residents in Singapore, recruiting two small groups of responses corresponding, respectively, to 49% and 39% of the people contacted and showing a reduction of perceived stress and perceived stigma levels compared to baseline [23]. Cai et al. observed a reduction in acute symptoms of stress recorded online by nurses in a large Chinese hospital in January and February 2020, corresponding to the attack and stationary phases of the pandemic [24]. Other researchers focused on sleep quality [25] and the effectiveness of cognitive behavioral therapy to prevent symptoms of stress [26]; on resilience [27,28]; on the relationship between leadership style and psychological safety and distress [29], and on the gradual adaptation of personnel to new safety procedures, with a reduction in anxiety and stress levels [30,31].

The summary of the prospective studies available so far shows that the evidence on the association between pandemic and mental health of HCWs is still limited. Repeated long-term cross-sectional studies are needed to determine how the mental health of frontline workers evolved during the different phases of the pandemic. These studies should consider several aspects of mental health at the same time, rather than focusing on a single psychological aspect or phenomenon.

In this article we report the results of a repeated cross-sectional study on the frontline workers of one of the two COVID-19 hub hospitals in Latium, Italy. The baseline survey, conducted during the first wave of the pandemic (April–May 2020), found that most workers reported high work-related stress; one out of three reported insomnia; one out of four experienced anxiety, and the majority reported depressive symptoms [32]. The workers expressed a modest level of confidence in their safety measures and reported a significant reduction in physical activity, meditation, and relaxation—the commonest ways of increasing resilience. Younger trainee workers (residents) expressed a significantly lower level of confidence in prevention measures than anesthesiologists, and this lack of organizational justice was associated with increased occupational stress [33].

Eight months after that study, when the second wave of infections was producing its effects, we set out to assess the condition of the workers who had continued to work in the hub hospital. This investigation was carried out before the start of the vaccination campaign. Our primary objectives were to assess the wellbeing and mental health of the workers after the first ten-month struggle with the virus and to evaluate the extent to which their attitude towards the pandemic had changed. Our ultimate aim was to identify the measures that would be most effective in preventing prolonged stress from impairing the health of workers.

## 2. Materials and Methods

### 2.1. Participants

All the workers in the anesthesiology department of the “A. Gemelli” University hospital in Rome (*n* = 205) who were directly involved in caring for suspected or confirmed cases of COVID-19 were emailed to invite them to participate in our survey. Their answers were collected anonymously on the SurveyMonkey online platform. Participation was completely voluntary and not economically incentivized. Participants were enrolled between 14 December 2020, and 5 January 2021. A reminder mail was sent twelve days after commencing the investigation.

Of the 205 eligible workers, 152 completed the survey (participation rate = 74.1%). Participants were mainly young (70.4% under 35 years of age), female (93, 61.2%) workers. Of those taking part, 105 were physicians and 47 were nurses (30.9%). About a quarter of the participants (34, 22.4%) had been working in the hospital for less than one year; about a quarter (42, 27.6%) had worked there for 1–3 years, and half of the participants (76, 50%) had been employed in the hospital for more than three years. Many of the workers who had responded to the first survey, conducted eight months earlier, had quit their jobs. Forty workers (26.3%) had not participated in the previous survey, while over half of the participants (87, 57.2%) confirmed that they had already answered at baseline. Most participants (80, 52.6%) reported unprotected exposure to SARS-CoV-2 patients (Table 1). Six of them reported having had a false-positive antigen test at the periodic screening all hospital workers undergo, and 26 (17.1%) had contracted COVID-19. Most of the workers who had contracted the infection had mild symptoms that did not require treatment (16, 61.5%) or were completely asymptomatic (7, 26.9%); only 3 had mild symptoms that required home treatment.

The survey was conducted in accordance with the Helsinki Declaration. The Catholic University Ethics Committee approved the study (ID 3292).

### 2.2. Questionnaire

The questionnaire used in this study consisted of a series of ad hoc questions, referring to the specific working conditions, obtained from the suggestions of a focus group composed of qualified anesthetists and from a panel of validated questionnaires. In this second survey, some changes in the ad hoc section were made in order to identify specific conditions of the second phase of the epidemic. To avoid the lengthening of compilation times, two of the standardized questionnaires already used in the baseline survey in extended form were administered in reduced form.

The questionnaire was composed of 43 questions divided into 6 sections. The average time required for completion was 5 min. The questionnaire contained a section (11 items) regarding socio-demographic factors that could influence the outcome, e.g., gender, age class, length of service, type of work, and accident status (unprotected contacts with COVID-19 cases, positive oropharyngeal tests, previous infection). The second section (10 items) investigated the main changes in occupation and lifestyle caused by the epidemic.

Work-related stress was measured with the Italian version [34] of the “Effort Reward Imbalance” (ERI) questionnaire [35,36]. Responses were graded on a 4-point Likert scale from 1 = “strongly disagree” to 4 = “strongly agree”. The effort subscale was based on three questions (e.g., “My job has become more and more demanding”); the total score ranged from 3 to 12. The reward subscale was based on seven questions (e.g., “I receive the respect I deserve from my superior or an equivalently qualified person”); consequently, this score ranged from 7 to 28. Internal consistency reliability scores of the two effort and reward sub-scales in this study were 0.751 and 0.820, respectively (good) [37]. The difference between efforts and results was measured as a weighted ratio of effort and reward. Values above one were considered indicative of distress.

The fourth section contained three questions on procedural justice (PJ) regarding the regularity of safety procedures. This was measured using the Italian version [38] of the Colquitt Scale [39,40,41] by means of 3 items (e.g., “Are these procedures error-free?”). Each question was answered according to a 5-point Likert scale, from 1 = “I totally disagree” to 5 = “I strongly agree”, thus producing a scale ranging from 3 to 15. In this study, the reliability of the questionnaire, measured by Cronbach’s alpha, was 0.665 (acceptable).

The next section, concerning positive and negative aspects of work, contained a question on job satisfaction, expressed on a 7-point Likert scale ranging from extremely dissatisfied to extremely satisfied, according to Warr et al. [42]; a question about happiness, on a 10-point scale, according to Abdel-Khalek [43]; a question on burnout, on a 6-point scale, according to West et al. [44], and finally a question regarding the possibility of leaving the hospital (yes/no).

Psychological symptoms were measured with the Italian version [45] of the “Goldberg Anxiety and Depression Scale” (GADS) [46], composed of 18 binary items on anxiety (9 items) and depression (9 items). Typical questions were: “Have you had difficulty relaxing?” for anxiety, and “Have you felt lethargic?” for depression. Persons with an anxiety score of 5 points or more, or a depression score of 2 or more, were classified as potentially anxious. In this study, the reliability of the GADS subscales, measured by Cronbach’s alpha, was 0.784 for anxiety and 0.693 for depression (acceptable).

Sleep quality was measured with the 2-item version of the “Sleep Condition Indicator” (SCI-02) [47,48], which aims to assess insomnia according to the Diagnostic Statistic Manual 5 (DSM5). Each question was graded on a 5-point Likert scale, ranging from 4 to 0. The final score ranged between 0 and 8, with higher values indicating better sleep quality. Cronbach’s alpha was 0.814 (good). A score of ≤4 revealed possible insomnia disorder.

### 2.3. Statistics

The variables were analyzed in descriptive terms, mean and standard deviation for continuous variables, and frequency for categorical variables. The results obtained in the second wave were compared with those of the first wave by Student’s t test for continuous variables, Mann Whitney’s test for ordinal variables, and chi square test for categorical variables. The comparison of the observations conducted at the baseline (T_0_) and at the follow-up (T_1_) was carried out both by including all the workers, and by excluding from T_1_ those workers who were not present at T_0_.

The role of pandemic-induced changes on occupational stress level was investigated using a stepwise linear regression model. The independent variables included in the model were gender, age, physical activity, meditation, procedural justice, workload, monotony, compassion fatigue, isolation at work, and social loneliness. The perceived effort–reward imbalance was set as a dependent variable. In the stepwise method based on the *p*-value of F, the model starts by entering the variable with the smallest *p*-value (PIN *p* < 0.05); it then adds the second strongest predictor with the smallest *p*-value for F and so on. Variables already in the equation are removed if their *p*-value becomes larger than the default limit (POUT *p* > 0.1) due to the inclusion of another variable.

The relationship between perceptions of justice and stress and outcomes was studied by simple linear regression analysis for continuous variables (sleep quality, anxiety, and depression) and logistic regression for binary variables (sleepless, anxious, depressed, satisfied, happy, with burnout, willing to leave work).

Analyses were performed using IBM/SPSS 26.0 (IBM Corporation, Armonk, NY, USA).

## 3. Results

When questioned about changes in their work due to the pandemic, workers reported a significant increase in workload and compassion fatigue resulting from the need to inform relatives of the death of a patient. These problems, which had already been described in the survey conducted during the first wave, were reported with a significantly increased frequency in the second wave (Table 2). The difference with what was observed at the baseline remains highly significant, even excluding from the calculation the workers who did not participate in the first survey. Most workers complained of having to work in isolation, and of being severely isolated in their social life (Table 2). Workers also reported a major change in public opinion between the first and second phases of the pandemic. Nine out of ten (87.4%) observed that in the first phase of the epidemic, between March and May 2020, people expressed appreciation and trust in HCWs, while in the current stage (December 2020), 91.0% believed HCWs were viewed less favorably than in the past.

As for the factors that increase resilience, free time spent on physical activity and meditation was reduced or seriously reduced for most of the participants (81.8% and 67.9% for physical and spiritual activities, respectively) (Table 2). About one-third of the participants (31.5%) did not believe that meditative prayer could lead to spiritual wellbeing or that it was an important part of life. A favorable attitude towards prayer had a weak correlation with occupational rewards (Spearman’s rho = 0.180, *p* = 0.03).

The perception of justice in safety procedures (range 3–15) yielded an average value of 7.90 ± 2.26, considerably lower than the maximum. The average value was located at the 53rd percentile. In the survey conducted during the first wave, the perception of justice was measured on a scale containing a larger number of items than was used in the second wave. In the first survey, the mean response was found to be in the 53rd percentile (Table 3). The two observations, therefore, do not register a change in the HCWs’ perception of organizational justice and correctness of security measures.

An analysis of the occupational stress perceived by workers indicated, on average, a prevalence of the effort made to work over the material and intangible rewards received. Effort (range 3–12) was on average 9.64 ± 1.66, i.e., about 80% of the maximum. Effort increased significantly in the second wave compared to the first (*p* < 0.001). Reward (range 7–28) was 16.01 ± 4.08, close to 57% of the maximum score; no significant increase was observed compared to previously recorded reward levels. Consequently, mean ERI was 1.54 ± 0.63, and the vast majority of workers (118, 83.1%) were in a state of distress. The situation confirmed and underlined what had been observed in the first phase, with a very significant statistical increase in occupational stress (Table 3).

The average quality of sleep, measured by the SCI02 (range 0–8), was low (5.22 ± 2.39, 65% of the maximum score), and a large number of workers (105, 73.9%) suffered from insomnia. The number of workers who could be classified as “suffering from insomnia” was substantially similar to the estimate made in the first wave. Mean anxiety and depression scores, measured with GADS, were moderately high for anxiety (3.39 ± 2.45), and particularly high for depression (2.74 ± 2.06). According to the diagnostic criteria of the questionnaire, 45 workers (31.3%) were likely to suffer from clinically relevant anxiety syndrome and 90 (62.5%) from depression. Compared to the first wave, there was a slight but not significant increase in mean anxiety scores, whereas the increase in the mean score and number of cases for depression was very significant (Table 2 and Table 3).

The mean score for occupational satisfaction, measured as indicated by Warr et al. on a scale ranging from extremely dissatisfied to extremely satisfied, was 4.05 ± 1.48, corresponding to “I am uncertain”. A total of 51.4% of the workers were moderately, very, or extremely satisfied. The average score for happiness in life, measured on a scale from 1 to 10, was 6.46 ± 1.97. When asked how often workers experienced burnout—an occupational condition characterized by emotional exhaustion, depersonalization and a low sense of personal achievement—the average response was 3.56 ± 1.58. This level corresponded to the “several times a month” option.

To the question “Have you thought about leaving this job?” 60 workers (42.6%) answered affirmatively.

The association between occupational changes brought about by the pandemic and work-related stress was studied by stepwise linear regression. The resulting model, which explained a significant share of the variability (R^2^ = 0.34), indicated that stress was dependent on lack of procedural justice, increased workload, isolation at work (having to work alone), and lack of time for meditation and relaxation. Gender and age were not included in the model. Similarly, monotony, compassion fatigue, social loneliness, and physical activity were excluded from the model (Table 4).

The relationship between perceived justice, occupational stress, and mental health outcomes was studied using simple linear regression models. Effort was a highly significant predictor of low sleep quality, anxiety, and depression (Table 5).

Effort significantly increased the odds of being diagnosed as insomniac, anxious, depressed, or burned out, and it significantly reduced the odds of being satisfied and happy. On the other hand, the perception of procedural justice had a protective effect on insomnia, while reward significantly increased the odds of work satisfaction. The intention to leave the hospital was significantly predicted by effort, while reward was a strong protective factor (Table 6).

## 4. Discussion

This study, which is the second cross-sectional survey on frontline workers in a COVID-19 hub hospital in Rome where the baseline interview took place during the first wave of the outbreak [4], illustrates how the mental health of these workers evolved in relation to the pandemic. The high workload, isolation at work, uncertainty about safety procedures, and the sharp reduction in the time devoted to meditation and relaxation have led to a significant increase in occupational stress, which for over 80% of workers is characterized by a discrepancy between the effort made to work and the material and immaterial rewards received as a result of work.

At ten months from the outbreak of the COVID epidemic, the share of workers who felt they had been overworked, had reduced time for meditation, and had suffered compassion fatigue significantly increased compared to baseline. This continual state of tension has led to a high rate of sleep and anxiety disorders and low levels of job satisfaction and happiness. Between the first and second wave of the pandemic, the most alarming factor is the increase in cases of depression. More than 60% of workers had a score exceeding the cut-off level, corresponding to a 50% chance of being diagnosed as “depressed” when examined by a specialist. Over 40% of workers considered quitting their job, and about a quarter of those who took part in the first survey no longer worked in the hospital.

Our study demonstrates the effect the COVID-19 pandemic has had on the mental health of intensivists. To the best of our knowledge, this is the first study that has used a long-lasting, repeated, cross-sectional design to evaluate changes in the level of mental health induced by the prolonged duration of the pandemic. This is also one of the few studies that has not focused on a specific parameter of mental health, but it has sought to assess the complex set of factors that include the perception of justice of safety procedures, specific stressors related to the pandemic (such as compassion fatigue), individual resources, and relaxation techniques to understand the variation in occupational stress and the consequences of sleep, anxiety, depression, burnout, job satisfaction, and happiness in life. We are convinced that our study can make a substantial contribution to the consolidation of evidence concerning the effects of the pandemic on the mental health of HCWs.

Some of the factors found to be associated with occupational stress in our survey were reported in previous research. For example, excessive workload [49], isolation or lack of support at work [50], the lack of procedural justice, or insufficient information about the outbreak and protective measures [51] were found to increase stress in frontline HCWs. Our research revealed that lack of time for meditation was strongly associated with stress. It is well known that meditation can significantly increase workers’ resilience [52], and meditative prayer has been used to prevent burnout in workers [53]. Mindfulness techniques have been used to support HCWs struggling with COVID-19 [54]. Moreover, in our study, a positive attitude towards prayer was associated with greater reward; this result is in agreement with the observation of a Taiwanese team, indicating the positive effect of religion on psychological resilience in HCWs during the pandemic [55].

The effects found in our cohort correspond to those reported by other researchers: insomnia [56], anxiety [11], burnout [16], reduced happiness [57], lack of job satisfaction, and turnover intention [58] are common in HCWs. Many of these outcomes remained constant in our cohort between the first and second wave, and it is not easy to understand whether they were a result of the pandemic or were already present in the population. However, we observed a significant increase in workload and compassion fatigue, and a further reduction in the time devoted to meditation and mental activities, between the first and second waves. These factors were accompanied by an increase in what were already very frequent cases of depression, especially among the younger sector of the population. In fact, this has now become the dominant condition, concerning 6 out of 10 HCWs.

The observations conducted led us to suggest and implement preventive measures. To reduce the workload, the first, elementary measure is the increase in the workforce; though it is difficult to have highly skilled people when the epidemic poses an immediate need. An emergency measure implemented, in this as in other hospitals, was to hire the trainees on a fixed-term contract. These young workers have made an essential contribution to therapeutic activities; however, although carefully prepared and specifically trained on the new security measures needed to deal with the pandemic and supported by the work of a team of experienced workers, they perceived an average level of informational justice lower than other workers, and this increased their occupational stress [11]. The sudden assumption of responsibility and the abrupt change in work habits were certainly very strong stressors. Experience shows that, out of pandemic, a training course should be provided, which should include capacity building on the management of major accidents and the development of specific skills, such as wearing safety clothing and working in it.

During a pandemic, personnel policy should become more flexible and play an important role in moderating occupational stress. When it is not possible to limit the effort required by a protracted emergency, rewards for workers must be increased. The rapidity of an epidemic makes it necessary to immediately create incentives for productivity and make work attractive; otherwise, if the inertia of bureaucracy prevails, there is the possibility of observing, as in this case, that many workers have left the company between the first and second wave, and many others intend to do so. The loss of skilled personnel is a real threat.

A specific effort must address the mental health of frontline workers. During the pandemic, the workers who requested it received free psychotherapeutic support. However, the use of this aid was only episodic and certainly late. The dramatic experience of the pandemic has taught us that, regardless of it, a continuous effort to promote health should be conducted in all major hospitals. Teaching sleep hygiene, developing relaxation techniques, promoting physical activity, and meditation and prayer should always be done. At critical moments, psychological support should be offered, especially to counter the compassion fatigue and interference with family and social life that epidemics pose.

The main strength of this study lies in its prospective design, the only one that enables us to record how the perception of stress and the mental health of workers have evolved in relation to the pandemic. The analysis of stress and mental health at various times of the pandemic and after its conclusion will help to disentangle the effect of the epidemic from that of other common stressors in health care activities. Several epidemiological studies based on repeated cross-sectional models have been announced or have published baseline reports [59,60,61,62,63]. This study is the first to document the mental health differences of HCWs at the start of the pandemic and after ten months, when the second wave had its effects and vaccines were not yet available. Another strong point is the fact that it has recruited intensivists from one of the two COVID hub centers in Central Italy. Many HCWs have been struggling with one or more COVID-19 patients and could be defined as “frontline workers”; the sample examined in this study, however, is composed of a small group of workers who worked exclusively and continuously with COVID-19 patients. The increased prevalence of depression symptoms we have observed is certainly an effect attributable to working conditions during the pandemic. The weakness inherent in the survey method is that no objective verification can be made of the reliability of the answers provided. The brevity of the questionnaire was a further limitation: it had to be very short because the frontline workers had very little time to devote to responding, and if they were interrupted during the survey, the system prevented them from continuing the compilation. For this reason, in this second survey, we adopted the short form of the procedural justice and sleep quality scales, in order to restrict compilation time to within 5–7 min without omitting to measure the variables of interest.

The protection of mental health in HCWs is of paramount importance for ensuring quality care [64,65]. We are convinced that preventive intervention is urgently required and that, in addition to individual support action aimed at increasing resilience, this should include a series of structural provisions designed to increase the workforce, optimize production flows, lower workloads, and provide greater rewards.

## 5. Conclusions

The long duration of the pandemic has exposed the frontline HCWs to an unprecedented strain. Excessive and prolonged workload, isolation, uncertainty about safety measures, lack of time for meditation have resulted in widespread distress and this in turn was associated with various signs of impaired mental health. The prospective nature of this study allowed us to demonstrate the increased prevalence of depression in intensivists, who are a key figure in the treatment of patients affected by COVID-19. A set of organizational and supportive measures are needed to increase the well-being of workers and the quality of care.

## Figures and Tables

**Table 1 ijerph-18-07316-t001:** Characteristics of the population.

Variable	1st Wave		2nd Wave	
	*N*	%	*N*	%
Gender, male	85	47.2	59	38.8
Age, <35 years	104	57.8	107	70.4
Physician	154	85.6	105	69.1
Reporting unprotected exposure to COVID-19 patients	46	25.6	80	52.6
Participated in the previous survey	-	-	87	57.2
Reporting a false-positive swab test	-	-	6	3.9
Reporting COVID-19 disease	-	-	26	17.1

**Table 2 ijerph-18-07316-t002:** Changes reported during the COVID-19 outbreak and prevalence of high stress, insomnia, anxiety, and depression during the 1st and 2nd waves.

Reported Effect	1st Wave		2nd Wave			2nd Wave *		
	*N*	%	*N*	%	*p*	*N*	%	*p*
Increased/greatly increased workload	94	52.2	126	88.1	0.000	91	85.8	0.000
The work became more repetitive and monotonous	54	30.6	43	30.1	0.589	34	32.1	0.485
More frequent need to inform of the death of a relative	56	36.7	88	61.6	0.000	65	61.3	0.000
Isolation at work	-	-	72	50.4		53	50.0	
Isolation in life	-	-	115	80.5		84	79.2	
Time for physical exercise was shorter/much shorter	141	88.3	117	81.8	0.099	84	79.2	0.395
Time for meditation was shorter/much shorter	87	48.3	97	67.9	0.000	67	63.2	0.003
High stress (effort/reward weighted ratio >1)	139	77.2	118	83.1	0.192	91	85.8	0.105
Insomniac (SCI08 score ≤16; SCI02 score ≤ 4)	79	43.9	57	40.1	0.499	64	60.4	0.561
Anxious (GADS anxiety score ≥5)	45	25.0	45	31.3	0.212	32	29.9	0.442
Depressed (GADS depression score ≥2)	89	49.4	90	62.5	0.019	68	63.6	0.028

SCI08 = Sleep Condition Indicator; SCI02 = Sleep Condition Indicator, short form, two items; GADS = Goldberg Anxiety and Depression Scale; * excluding workers who were not present at T_0_.

**Table 3 ijerph-18-07316-t003:** Mental health indicators (perceived justice, occupational stress, sleep quality, anxiety, depression) in anesthesiologists during the 1st and 2nd waves of the COVID-19 outbreak.

Variable	1st Wave	2nd Wave		2nd Wave *	
	Mean ± s.d. (% max)	Mean ± s.d. (% max)	*p*	Mean ± s.d. (% max)	*p*
Procedural Justice **	31.8 ± 7.3 (52.8%)	7.9 ± 2.3 (52.7%)		7.8 ± 2.4 (52.0%)	
Effort	8.6 ± 1.9 (71.7%)	9.6 ± 1.7 (80.0%)	0.000	9.6 ± 1.6 (80.0%)	
Reward	16.5 ± 3.6 (58.9%)	16.0 ± 4.1 (57.1%)	0.251	16.0 ± 4.2 (57.1%)	
Job stress	1.31 ± 0.49	1.54 ± 0.63	0.000	1.54 ± 0.61	0.001
Sleep quality *	21.2 ± 8.2 (66.2%)	4.78 ± 2.38 (59.7%)		4.91 ± 2.44 (61.3%)	
Anxiety	3.04 ± 2.29	3.39 ± 2.45	0.193	3.39 ± 2.35	0.222
Depression	1.98 ± 1.82	2.74 ± 2.06	0.000	2.58 ± 1.78	0.007

* Excluding workers who were not present at T_0_; ** the questionnaire was administered in a reduced form in the 2nd survey.

**Table 4 ijerph-18-07316-t004:** Second wave. Stepwise linear regression analysis. Relationship between job changes and perceived work-related stress (ERI).

Variable	ERI	
	Standardized Beta	*p*
Procedural Justice	−0.310	0.001
Workload	0.270	0.001
Isolation at work	0.199	0.007
Meditation	−0.151	0.034
Determination coefficient of the model (R^2^)		0.343

Variables excluded from the model: gender, age class, monotony, compassion fatigue, social loneliness, physical activity.

**Table 5 ijerph-18-07316-t005:** Second wave. Health outcomes associated with procedural justice and occupational stress. Linear regression analysis adjusted for age and gender.

Variable	Sleep Quality		Anxiety		Depression	
	Standardized Beta	*p*	Standardized Beta	*p*	Standardized Beta	*p*
Procedural justice	0.169	0.062	−0.110	0.220	0.065	0.453
Effort	−0.349	0.000	0.345	0.000	0.364	0.000
Reward	0.059	0.521	−0.067	0.464	−0.151	0.091

**Table 6 ijerph-18-07316-t006:** Second wave. Health outcomes associated with procedural justice and occupational stress. Multivariate logistic regression model adjusted for age and gender.

**Variable**	**Insomniac ^1^**		**Anxious ^2^**		**Depressed ^3^**	
	**OR (CI95%)**	***p***	**OR (CI95%)**	***p***	**OR (CI95%)**	***p***
Procedural justice	**0.794** (0.646, 0.977)	0.029	0.950 (0.766, 1.177)	0.639	0.980 (0.805, 1.193)	0.838
Effort	**1.780** (1.342, 2.360)	0.000	**1.858** (1.360, 2.539)	0.000	**1.363** (1.063, 1.748)	0.015
Reward	1.053 (0.942, 1.178	0.365	0.951 (0.841, 1.075)	0.421	0.956 (0.857, 1.067)	0.420
**Variable**	**Satisfied ^4^**		**Happy ^5^**		**Burned out ^5^**	
	**OR (CI95%)**	***p***	**OR (CI95%)**	***p***	**OR (CI95%)**	***p***
Procedural justice	1.006 (0.806–1.257)	0.955	0.911 (0.755, 1.100)	0.334	1.083 (0.873, 1.344)	0.467
Effort	**0.515** (0.375, 0.706)	0.000	**0.771** (0.604, 0.983)	0.036	**1.918** (1.398, 2.631)	0.000
Reward	**1.277** (1.115, 1.463)	0.000	1.103 (0.991, 1.227)	0.073	0.940 (0.834, 1.060)	0.312
**Variable**	**Intention to leave**					
	**OR (CI95%)**	***p***				
Procedural justice	1.049 (0.859, 1.281)	0.639				
Effort	**1.413** (1.077, 1.852)	0.012				
Reward	**0.****792** (0.699, 0.896)	0.000				

Notes: ^1^ = SCI02 score ≤4; ^2^ = GADS anxiety score ≥5; ^3^ = GADS depression score ≥2; ^4^ = moderately, very, or extremely satisfied; ^5^ = dichotomized at the median.

## Data Availability

Data deposited on Zenodo: DOI 10.5281/zenodo.5081625.

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
