# Peer review of "Prolonged Stress Causes Depression in Frontline Workers Facing the COVID-19 Pandemic—A Repeated Cross-Sectional Study in a COVID-19 Hub-Hospital in Central Italy"

_ijerph, 2021, doi:10.3390/ijerph18147316_

Round 1

Reviewer 1 Report

Great job

Reviewer 2 Report

The corrected article presented for review touches on a very current research problem. The content of the thesis corresponds to the topic. The abstract is concise and contains all the necessary elements. The content presented in the theoretical introduction fully introduces the research topic and sufficiently justifies the main problem of the work. The aim of the work was formulated correctly. The methodology is correct. The research results are presented in a legible, transparent and comprehensive way.

This manuscript is a resubmission of an earlier submission. The following is a list of the peer review reports and author responses from that submission.

Round 1

Reviewer 1 Report

Comments:

I hope all these suggestions will not discourage the author(s) keep working on this manuscript in the future. Good luck to the author(s).

Additional Questions:

  1. Originality: The study of Prolonged stress causes depression in frontline workers facing the COVID-19 pandemic. A repetead cross-sectional sutdy significantly contributes to the understanding of trends in frontline workers and relationships between employees and organizations and expands our scientific perspective. In that sense, the research context of this paper is very interesting. In addition, the latest figures highlight how the prolongued srtess evolved during the pandemic. Thus, I believe further research in this topic is needed. Nevertheless, there are some relevant concerns in this paper which are shown below.

  1. Relationship to Literature In general terms, the paper do not demostrate an understanding of the relevant literature in the mental stress and labour market field. Thus, I think the literature review is a weak in this paper. However, there are some relevant concerns in this section that should be improved:

  • Some important statements should be supported by the literature. So, for example, on line 51, it is said verbatim “Overall, a lack of longitudinal studies made it impossible to distinguish the effect of the pandemic from that of all other pre-existing stressors in hospital work, and there is still little evidence of an increase in mental health problems in HCWs during the outbreak”

  • In the introduction in line 54 “A repeated cross-sectional study 54 will therefore help to determine how the mental health of frontline workers evolved 55 during the different phases of the pandemic. I think this is the gap of the research but I think you should explain deeply, because I think this is the main point of this article which is your main contribution and it must be something more important than this one

  • The next section is methodology and in my opinion it is necessary a literature review of the topic. I have seen that the article has not this important part of the article, which must present. A literature review is a search and evaluation of the available literature in your given subject or chosen topic area. It documents the state of the art with respect to the subject or topic you are writing about. The literatura review: demonstrates a familiarity with a body of knowledge and establishes the credibility of your work; summarises prior research and says how your project is linked to it; integrates and summarises what is known about a subject; demonstrates that you have learnt from others and that your research is a starting point for new ideas.

  1. Implications for research, practice and/or society: There is not a clear appart for conclusions. The author does not explain what the theoretical contributions are or the practical implications for the topic. Neither makes proposals or recommendations. The conclusions are not solidly based in the empirical obtained results. Conclusions are merely descriptives.

Reviewer 2 Report

Taking up the research topic by the Authors deserves recognition. Unfortunately, this project was based on incorrect methodological assumptions. This is not a longitudinal study as it is not the same group of respondents in the first and second studies. The number of people who participated in both studies is less than 50%.
In my opinion, the longitudinal test criterion has not been met.
It is worth describing these two groups separately due to the waves of the pandemic.

I propose to analyze the results of people who participated in both studies as a longitudinal study.